# Improving Classification Performance of Softmax Loss Function Based on Scalable Batch-Normalization

**Qiuyu Zhu [1,\*], Zikuang He [1] , Tao Zhang [2] and Wennan Cui [2]**

[1]  School of Communication & Information Engineering, Shanghai University, Shanghai 200444, China; hezikuang@shu.edu.cn

[2]  Key Laboratory of intelligent infrared perception, Chinese Academy of Sciences, Shanghai 200083, China; sitp_710@mail.sitp.ac.cn (T.Z.); cuiwennan@mail.sitp.ac.cn (W.C.)

\*  Correspondence: zhuqiuyu@staff.shu.edu.cn; Tel.: +86-130-0311-5119



**Featured Application: This work can be widely used in all kinds of pattern recognition systems based on deep learning, such as face recognition, license plate recognition, and speech recognition, etc.**

**Abstract:** Convolutional neural networks (CNNs) have made great achievements on computer vision tasks, especially the image classification. With the improvement of network structure and loss functions, the performance of image classification is getting higher and higher. The classic Softmax + cross-entropy loss has been the norm for training neural networks for years, which is calculated from the output probability of the ground-truth class. Then the network's weight is updated by gradient calculation of the loss. However, after several epochs of training, the back-propagation errors usually become almost negligible. For the above considerations, we proposed that batch normalization with adjustable scale could be added after network output to alleviate the problem of vanishing gradient problem in deep learning. The experimental results show that our method can significantly improve the final classification accuracy on different network structures, and is also better than many other improved classification Loss.

**Keywords:** convolutional neural network; loss function; gradient decent

## 1. Introduction

Recently, deep convolutional neural networks have made great achievements in many aspects of computer vision tasks. Image classification [1] is a fundamental task and full of challenges, thanks to the large number of labeled training data, supervised learning has reached a level beyond human beings. The current convolutional neural network training process for image classification is shown in Figure 1, which includes several modules of feature extraction, feature representation, category prediction, and loss calculation.

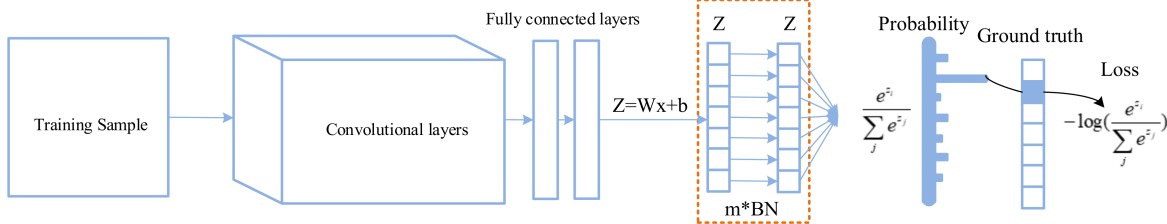

**Figure 1.** Traditional training process for classification tasks and new scaled Batch Normalization (BN) layer.

As shown in Figure 1, given a training sample, features of the input sample are extracted through a series of convolution and pooling layers. Then, some linear layers and Softmax classifier are used to determine the prediction probability that the input image belongs to each category. Finally, the loss at this epoch is obtained by calculating the cross-entropy loss, the network's parameter is optimized by back propagation algorithm to minimize the loss of network output.

With the deepening of research, many well-designed network structures have been proposed to improve the accuracy in image classification task, such as VGG (Visual Geometry Group)[2], ResNet [1], ResNeXt [3], DenseNet [4]. Their contribution mainly based on the network structure's improvement, the method of multi-layer feature concatenation has solved the problem of overfitting and gradient vanishing which have puzzled the researchers for a long time to some extent. In addition to the optimization of network structure, the optimization of loss function has also gained much attention in recent years. As a classical classification problem loss function, Softmax+cross-entropy loss has made great achievements, and its calculation method is as follows:

$$L(y, \hat{y}) = -\frac{1}{N}\sum_{i=1}^{N} y_i^T log(\hat{y}_i) = -\frac{1}{N}\sum_{i=1}^{N} log(\hat{y}_{ig}) \tag{1}$$

where $y_i \in \{0,1\}^k$ represents the label of the $i^{th}$ sample in one-hot encoded representation and $\hat{y}_i \in [0,1]^k$ is the predicted probabilities with Softmax. $\hat{y}_{ig}$ represents the predicted probability of the ground-truth class for the $i^{th}$ sample. Therefore, the network parameters can achieve $\hat{\theta} = argmin_\theta L(y, \hat{y})$ by training the cross-entropy loss function, where $\hat{y} = h_\theta(x)$, $h_\theta$ is the network's parameter and $x$ is the training sample. Although the Softmax+cross-entropy loss function has achieved great success in many computer vision fields as a norm of neural network, some improvements have been made by researchers to let the features learned by the convolutional neural network more distinctive between classes. Loss functions such as center-loss [5], L-Softmax [6], A-Softmax [7], AM-Softmax (Additive Margin) [8], PEDCC-loss (Pre-defined Evenly-Distributed Class Centroids) [9], etc. can improve the final accuracy of models when trained for face classification and validation.

Although the accuracy of the convolutional neural network for classification and recognition is constantly improving through the improvement of model structure and the optimization of loss function, there are still some unsolved problems. At present, various improved loss functions are proposed from the perspective of the output feature of convolutional neural network, that is, how to make the features extracted by the neural network have smaller intra-class distance and larger inter-class distance. The PEDCC-loss which is recently proposed achieves the best performance in these optimization methods, which are normally motivated from the perspective of face recognition to improve the accuracy of network classification. The problem is not considered from the point of view of solving the gradient vanishing problem and they did not test on the datasets which requires good generalization ability. For clarity, we summarize our research and contributions as follows:

- From the point of view of solving the problem of gradient vanishing, we propose to add a scaled Batch Normalization (BN) layer before Softmax computing to solve the problem during the training stage of convolutional neural network, let the network's parameter can be optimized continuously.
- We propose to add a scalable Batch Normalization (BN) layer before Softmax computing to improve the performance of Softmax+cross-entropy loss, which can boost the final classification accuracy of the convolutional neural network.
- The experimental results show that the final accuracy of network is improved in different data sets and different network structures compared with existing loss functions.

## 2. Related Work

### 2.1. Loss Functions in Classification Supervised Learning

As a common loss function in the training of classification task, the cross-entropy loss plays an important role in the training process of the neural network to measure whether the current model is good enough or not. Loss calculated based on this criterion can update the model parameters by its gradient, and the output loss could be minimized by this way. Currently improved classification loss functions are usually extended from the standard cross-entropy loss, such as L-Softmax and AM-Softmax. L-Softmax is the first method to improve the cross-entropy loss by placing an angular decision boundary between different classes to learn more discriminative features. AM-Softmax is an improvement on L-Softmax, which focus on the feature normalization and weight normalization. The cross-entropy loss function is optimized to make the features extracted from the neural network more representative. For instance, the calculation formula of AM-Softmax is as follows:

$$L_{AM-Softmax} = -\frac{1}{N}\sum_i log \frac{e^{s \cdot (cos\theta_{y_i} - m)}}{e^{s \cdot (cos\theta_{y_i} - m)} + \sum_{j=1, j \neq y_i}^{c} e^{s \cdot cos\theta_j}} \tag{2}$$

where $m$ and $s$ are adjustable super parameters, whose main idea is altering the original linear layer's formula $W \cdot x$ to $\|W\| \cdot \|x\| \cos(\theta)$ according to law of cosines. By means of feature normalization and linear layer weight normalization, the category prediction of the input image is only determined by the cosine distance between the image features extracted by the convolutional neural network and the corresponding normalized weight.

Through the adjustment of super parameters, the features learned by the neural network could have smaller intra-class distance and larger inter-class distance. Moreover, feature normalization in face recognition task can make neural network pay more attention to low-quality training samples. However, in the use of other datasets, AM-Softmax does not perform as well as expected. In addition to face recognition, common image classification data sets, such as CIFAR100 [10] and TinyImagenet [11], often appear the problem of untrainable due to feature normalization. Besides, it has two super parameters, so it is necessary to continuously adjust parameters according to the training results, which increases the difficulty of training process. In the data set that requires higher generalization, this improvement effect on the final accuracy is also very limited.

### 2.2. Batch Normalization

Normalization is one of the most important methods in neural network training phase. For example, normalization method is used to preprocess the training dataset. Experience has proved that training will be more stable and convergent if each channel of RGB image is normalized [12]. In the deep neural network, except that the distribution of input data in the first layer is consistent because the input is training data set, the distribution of input data in other hidden layers is not fixed. Because when the network updates the parameters it will inevitably have an impact on the output feature distribution of each layer, and the input of each layer will be affected by the parameters of all previous layers. Therefore, even if the change of parameters is small, the influence will be amplified as the network becomes deeper. This makes it very complicated to train a network, also called internal covariate shift problem. The basic idea of BatchNormalization [13] is to fix the input distribution for each hidden layer so that the network can train at a higher learning rate.

In addition, the operation of BatchNormalization after each convolutional layer of the convolutional neural network brings another benefit. The feature distribution of each hidden layer in the network always follows the change of parameters, and the distribution gradually approaches the upper and lower limits of the value interval of activation function. Some activation functions like sigmoid are affected by this, resulting in the gradient disappearance during the back propagation. BatchNormalization can let the distribution of output feature falls in the range with larger gradient of activation function.

It also means that the learning convergence speed is faster, so that the speed and stability of training can be greatly improved. The formula is as follows:

$$\hat{x}^{(k)} = \frac{x^{(k)} - E\left[x^{(k)}\right]}{\sqrt{Var\left[x^{(k)}\right]}} \tag{3}$$

where $x^{(k)}$ represents the output for the previous layer of each channel. After this transformation, the output of each layer will form a normal distribution with a mean of 0 and a variance of 1, but this alone will lead to a decline in the network's ability to fit the data set. In order to obtain nonlinear information, BatchNormalization adds two learnable parameters (scale and shift), the actual operation process is as follows:

$$y^{(k)} = \gamma^{(k)}\hat{x}^{(k)} + \beta^{(k)} \tag{4}$$

## 3. Inserting Scalable BN Layer to Improve the Classification Performance of Softmax Loss Function

Through the observation of Softmax and cross-entropy loss, we found that the calculation of cross-entropy loss mainly depends on the corresponding probability of the network output of the correct category through Softmax. The Softmax function is as follows:

$$p_i = \frac{e^{z_i}}{e^{z_i} + \sum_{j=1, j\neq i}^{N} e^{z_j}} \tag{5}$$

where $z_i = W \cdot x$ is the $i$-d output of the last linear layer of the convolutional neural network, $W$ is the linear layer's weight, $x$ is the output feature from the layer before the linear layer. $p_i$ is the result of $z_i$ after Softmax calculation, it converts the output of the linear layer into a probabilistic representation.

After observing the formula of Softmax, we found that if we want to increase the prediction probability of the image's correct category, one of the simplest methods is to multiply the value of linear layer weight by a constant value greater than 1. However, this way of improving the accuracy of the final prediction cannot enhance the feature extraction ability of the convolutional neural network, and is useless for improving classification performance. The key of the problem is to let the linear layer's input feature be more effective. In our experiment, we will display the contrast diagram.

Let us take a classification problem with three categories as an example. One image sample belongs to the third category pass through the neural network and its final feature vector is [1,1,2]. Then, the probability that it belongs to the correct category calculated by Softmax is $e^2/\left(e^1 + e^1 + e^2\right) = 0.6769$. If the final linear layer weight $W$ is simply multiplied by 2, then the final output feature of the image will become [2,2,4], and the prediction probability increases to $e^4/\left(e^2 + e^2 + e^4\right) = 0.8481$. Without improving the ability of neural networks to extract features, the prediction accuracy of the image was improved by 17.12%. This will undoubtedly affect the effectiveness of updating network weights by back propagation through the cross-entropy loss, that is because the gradient disappears, the loss function of the network lost its ability to update the parameters in the network. It will also be demonstrated in the following experiments.

The updating gradient is calculated by taking the derivative of the final output vector of the network with cross-entropy loss, which can be illustrated by $\partial L/\partial Z_j$:

$$\frac{\partial L}{\partial Z_j} = -\sum_{i=1}^{N} y_i \frac{\partial \log(p_i)}{\partial Z_j} = -\sum_{i=1}^{N} \frac{y_i}{p_i} \frac{\partial p_i}{\partial Z_j} \tag{6}$$

where $p$ is the output of $z$ after the softmax function, $p = \text{softmax}(z)$ the solution of this can be divided into two cases, one is $i = j$ and another is $i \neq j$:

$$\frac{\partial L}{\partial Z_j} = -\frac{y_j}{p_j}p_j\left(1 - p_j\right) - \sum_{i \neq j}\frac{y_j}{p_j}\left(-p_i p_j\right) = p_j - y_j \tag{7}$$

In the vector form:

$$\frac{\partial L}{\partial z} = \boldsymbol{p} - \boldsymbol{y} \tag{8}$$

It can be seen that when the output prediction vector of the neural network is closer to the correct label, the gradient update amplitude of the back propagation will be lower and lower, resulting in the phenomenon of gradient vanishing problem, which will greatly affect the effectiveness of feature extraction.

In order to maintain the ability of gradient updating in the training process of neural network, researchers also made some explorations, such as normalizing the weight of linear layer and the input feature, which also alleviated this problem to some extent, but the effect is less obvious in general data sets. In this paper, the idea of—Batch Normalization is adopted to normalize the channel at the final output of the neural network, the cross-entropy loss is then calculated to update the network parameters. However, without adding learnable parameters, the output distribution of the neural network is artificially controlled by a super parameter $m$:

$$z^{(k)} = m\frac{z^{(k)} - E\left[z^{(k)}\right]}{\sqrt{Var\left[z^{(k)}\right]}} \tag{9}$$

When the network adopts Resnet18 and the data set is CIFAR100, the distribution of the final network output before and after inserting the batch normalization layer, and the modulus comparison of the corresponding weight of each class of linear layer are shown in Figures 2 and 3, where the abscissa axis of Figure 2 is the output amplitude of the neural network, and the ordinate axis is the frequency of the occurrence of the amplitude. It is obvious from the comparison of Figure 2 that the output value of the traditional neural network training method is concentrated between −5 and 5. After the inserting of batch normalization, the final network output values are concentrated between −2 and 2. It also can be seen from Figure 3 that the corresponding weight module of each class in the last linear layer is much smaller than that of the traditional training method. Thus, combining with the aforementioned gradient update formula, it can be seen that the back propagation gradient of network parameters is significantly strengthened at this time, and the problem of gradient vanishing caused by the increase of linear layer weight is also solved.

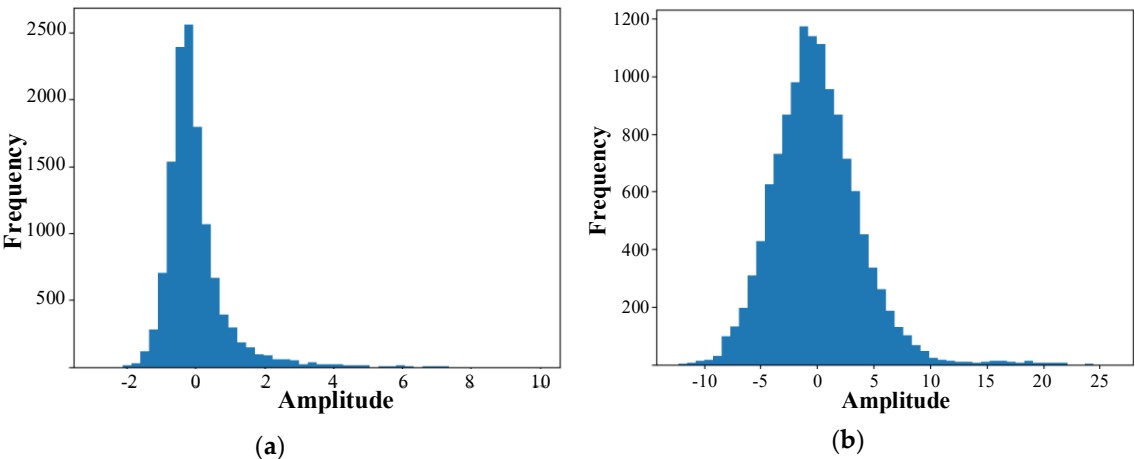

(a)

(b)

**Figure 2.** Left picture (**a**) is the network's output distribution after the traditional training process, the right figure (**b**) shows the output distribution with the BN constraint.

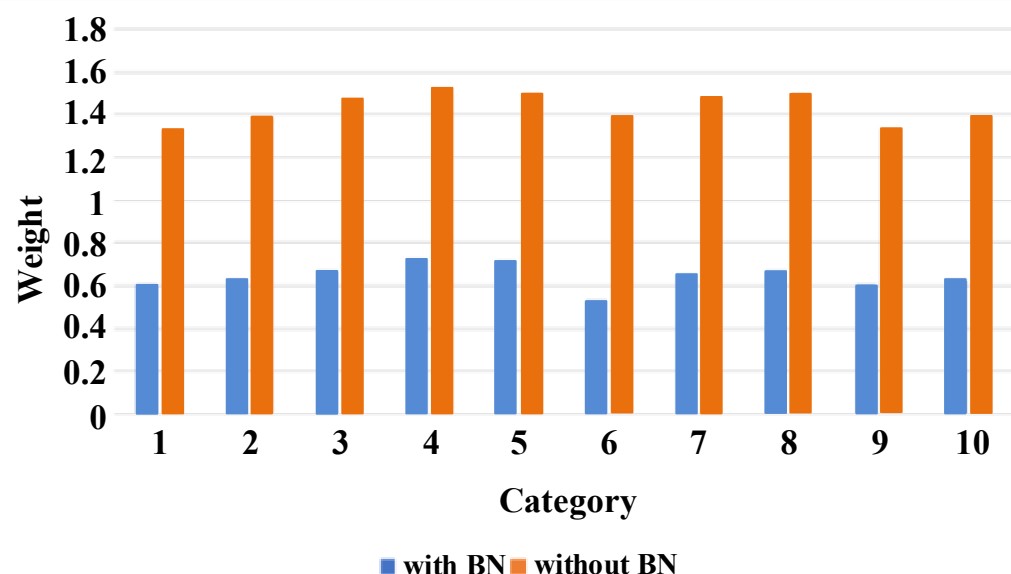

**Figure 3.** Modulus comparison of the first ten classes of linear layer corresponding to weights.

## 4. Experiments and Results

In this section, we compare the training strategy proposed in this paper with cross-entropy Loss and the most advanced AM-softmax Loss, PEDCC Loss on three widely used data sets with many categories to show the advantages of our method. We also cover several different network structures which prevalently used recently to demonstrate the robustness of our method. All experiments are built on CNN tools framework: PyTorch [14]. The computing server is equipped with a single Nvidia GTX 1080Ti GPU. In the training phase of all experiments, we set the batch size to 128, and adopt SGD as the optimization algorithm. The momentum is set to 0.9, and the weight decay is 0.0005. The initial learning rate is set to 0.1, which will be multiplied by 0.1 at the 30th, 60th, and 90th epoch. The maximum number of epochs is 100.

In order to prove the effectiveness of our method, we choose the most widely used framework VGG and ResNet as the basic network structure. Three data sets, namely CIFAR100, TinyImagenet and Facescrub, which require high generalization performance were evaluated. CIFAR100 has 100 classes, image resolution is $32 \times 32$, and there are 500 training images and 100 test images for each class. TinyImagenet has 200 classes, image resolution is $64 \times 64$, and the number of training and test images for each class is 500 and 50, respectively. In the Facescrub dataset, we eliminated categories with less than 70 samples. For each category, 10% of the images are randomly selected as the test set and the image are resized to $64 \times 64$. Finally, we have 361 categories, each one has 90 images to train and 10 images to test. The final results are shown in Tables 1–3, respectively. All experimental results show that our method has better classification accuracy than classic Softmax loss, AM-Softmax, and PEDCC-loss. Here we set two sets hyperparameter $s = 7.5$, $m = 0.35$, and $s = 10$, $m = 0.5$, which is following the original PEDCC-loss [9]. The hyperparameter $s$ is used to let the network be trainable. As the features and every column of weight are normalized, the output range is only $[-1, 1]$, which is not suitable for Softmax loss [15]. Another hyperparameter $m$ represents the expected cosine margin between each class after training [8]. For each run, we repeat 3 times and report the "mean(±std)" values.

**Table 1.** The experimental results of CIFAR-100 dataset.

| CIFAR-100 Results | VGG | ResNet-18 | ResNet-50 |
|---|---|---|---|
| Cross-entropy Loss | 71.86% (±0.14%) | 75.10% (±0.34%) | 75.95% (±0.32%) |
| AM-SoftMax Loss (s = 7.5, m = 0.35) | 72.85% (±0.18%) | 74.97% (±0.08%) | 75.68% (±0.14%) |
| AM-SoftMax Loss (s = 10, m = 0.5) | 71.97% (±0.54%) | 74.40% (±0.38%) | 75.61% (±0.24%) |
| PEDCC-Loss (s = 7.5, m = 0.35) | 72.53% (±0.32%) | 74.86% (±0.51%) | 76.19% (±0.14%) |
| PEDCC-Loss (s = 10, m = 0.5) | 71.98% (±0.31%) | 73.84% (±0.48%) | 75.41% (±0.17%) |
| Our Loss (m = 0.8) | 73.57% (±0.25%) | **76.12% (±0.31%)** | 76.56% (±0.13%) |
| Our Loss (m = 0.9) | **74.13% (±0.24%)** | 76.10% (±0.12%) | **76.62% (±0.17%)** |
| Our Loss (m = 1) | 73.24% (±0.06%) | 76.08% (±0.30%) | 76.32% (±0.51%) |

**Table 2.** The experimental results of TinyImagenet dataset.

| TinyImagenet Results | VGG | ResNet-18 | ResNet-50 |
|---|---|---|---|
| Cross-entropy Loss | 54.05% (±0.19%) | 59.38% (±0.18%) | 62.35% (±0.27%) |
| AM-SoftMax Loss (s = 7.5, m = 0.35) | 54.57% (±0.28%) | 59.48% (±0.25%) | 61.83% (±0.27%) |
| AM-SoftMax Loss (s = 10, m = 0.5) | 54.62% (±0.31%) | 59.26% (±0.32%) | 62.40% (±0.06%) |
| PEDCC-Loss (s = 7.5, m = 0.35) | 54.91% (±0.17%) | 59.13% (±0.38%) | 62.38% (±0.52%) |
| PEDCC-Loss (s = 10, m = 0.5) | 54.22% (±0.21%) | 59.04% (±0.08%) | 6242% (±0.24%) |
| Our Loss (m = 0.8) | 55.02% (±0.09%) | 60.34% (±0.32%) | **63.72% (±0.08%)** |
| Our Loss (m = 0.9) | 55.13% (±0.25%) | 60.12% (±0.07%) | 63.09% (±0.15%) |
| Our Loss (m = 1) | **55.86% (±0.26%)** | **60.61% (±0.23%)** | 63.27% (±0.30%) |

**Table 3.** The experimental results of Facescrub dataset.

| Facescrub Results | VGG | ResNet-18 | ResNet-50 |
|---|---|---|---|
| Cross-entropy Loss | 93.54% (±0.30%) | 89.65% (±0.13%) | 90.01% (±0.31%) |
| AM-SoftMax Loss (s = 7.5, m = 0.35) | 94.12% (±0.08%) | 91.24% (±0.24%) | 90.84% (±0.35%) |
| AM-SoftMax Loss (s = 10, m = 0.5) | 93.95% (±0.19%) | 91.01% (±0.29%) | 91.67% (±0.50%) |
| PEDCC-Loss (s = 7.5, m = 0.35) | 93.68% (±0.37%) | 90.50% (±0.25%) | 90.47% (±0.34%) |
| PEDCC-Loss (s = 10, m = 0.5) | 93.82% (±0.27%) | 90.27% (±0.30%) | 90.98% (±0.36%) |
| Our Loss (m = 0.8) | **94.73% (±0.16%)** | 92.32% (±0.16%) | 92.28% (±0.46%) |
| Our Loss (m = 0.9) | 94.36% (±0.08%) | **92.47% (±0.09%)** | 92.00% (±0.61%) |
| Our Loss (m = 1) | 94.44% (±0.37%) | 92.36% (±0.30%) | **92.38% (±0.14%)** |

## 5. Conclusions

Based on the influence analysis of the amplitude of classification features and weight of the last linear layer on the back propagation gradient, this paper proposes a BN based new classification approach. Different from the conventional idea of optimization of the cosine distance of Softmax loss, we try to alleviate the gradient vanishing problem by using the scalable batch normalization layer to achieve our goal and it makes the extracted features of the network more effective. The experimental results also prove that the proposed method can improve significantly the accuracy of the network classification in different data sets and different network structures, and is also better than many other improved classification losses. In future research, we could explore a new loss function whose gradient is better than cross-entropy loss.

**Author Contributions:** Data curation, T.Z. and W.C.; Funding acquisition, T.Z. and W.C.; Methodology, Q.Z.; Software, Z.H.; Supervision, Q.Z.; Writing—original draft, Z.H. All authors have read and agreed to the published version of the manuscript.

**Funding:** This research was funded by the open project fund of the Key Laboratory of intelligent infrared perception, Chinese Academy of Sciences (grant number:CAS-IIRP-2030-03), and the APC was funded by CAS-IIRP-2030-03.

**Conflicts of Interest:** The authors declare no conflict of interest.

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
