# Peer review of "Improving Classification Performance of Softmax Loss Function Based on Scalable Batch-Normalization"

_applsci, doi:10.3390/app10082950_

Round 1
Reviewer 1 Report
This work proposes batch normalization with adjustable scaling after network output to alleviate the problem of vanishing gradient problem in deep learning
The experimental results show the proposed method improves the final classification accuracy on different network structures.
Literature needs to be improved and needs to compare with state of the art in the form of a table.
All equations taken from literature need to be referenced properly.
The batch normalization section need to be enhanced and must be explained further.
Quality of Figures 2 and 3 need to be improved and font must be enlarged.
Manuscript need to be double-checked for typos as well.
Author Response
First of all, thanks a lot for your very careful review of our paper.Your comments mean a lot to us. The changes are shown in "Track Changes" in a new version of our manuscript . 1. Response to comment: (literature needs to be improved and needs to
compare with SOTA in the table)
Response: We made some modification to the paper, and added the statistical
analyses in the result table, the state of art method which we compare with is
AM-Softmax loss and PEDCC-loss.
2. Response to comment: (equations need to be referenced properly)
Response: We are very sorry for our negligence of ignoring the reference to an equation (line 169), which caused problems with subsequent references. Thank you for your careful review, the changes had been made.
3. Response to comment: (batch normalization section need to be enhanced and explained further)
Response: We re-phrased the batch normalization section (2.2), explained more about why batch normalization can improve neural network’ training phase. Besides, we changed some informal sentences, added a reference to support some statement and re-wrote the long sentence to make it clearer.
4. Response to comment: (quality of Fig 2 and 3 need to be improved and font must be enlarged)
Response: We are so sorry that the picture does look blurry, especially the text on it. We modified the Figure, enlarged the fonts on it to make it clear.
5. Response to comment: (Manuscript need to be double-checked for typos)
Response: We are very sorry for our incorrect writing. We found that some words and phrase are inappropriate, and made some modification on the typo problem such as changing the bold text to no-bold text and modifying some verb tenses.
Finally, special thanks to you for your time and good comments. That means a lot to us.
Reviewer 2 Report
The paper analyses a common problem of the ANN training and proposes an alternative way to tackle it. IMO, the analysis of the state-of-the-art is quite complete. The solution proposed is evaluated under three different architectures and three datasets so it has a sound scientific basis. However, there are a few points that must be improved:
- One of the important missing points is a proper justification for the results. Even though the improvement regarding the classification performance is clear, statistical analyses must be conducted to support the conclusions
- The organization of the sections is also misleading. For example, the introduction and the related work sections seem to overlap in some parts. Most importantly, the "Experimental Results" section is not only including results but also the description of the dataset, the methodology, etc.
- Additionally, the conclusion of the work is mentioned in the related work section (lines 185-187) which also include the description of some experiments
- The configurations evaluated in the table to compare results should be also discussed (why these ones? why these parameters, how they affect the results? etc)
- There are a few parts to be re-phrased to provide more detailed and/or clear descriptions (see the attached file)
- There are several typos and language-related errors are found throughout the paper (see a few examples in the attached document
More concrete details are included in the attached document. Please revise the comments there

Author Response
First of all, thanks a lot for your very careful review of our paper. Your comments mean a lot to us.
The changes are shown in "Track Changes" in a new version of our manuscript .
- Response to comment: (statistical analyses must be conducted to support the conclusions)
Response: We added the statistical analyses in the result table. For each run, we repeat 3 times and report the “mean(±std)” values. - Response to comment: (The organization of the sections is also misleading)
Response: We are very sorry for our inappropriate writing. We deleted and re-wrote some overlap parts in different sections, and changed the misleading title “Experimental Results” to “Experiments and results”. - Response to comment: (the conclusion of the work is mentioned in the related work section (lines 185-187) which also include the description of some experiments)
Response: Sorry for our inappropriate writing again. We deleted the unsuitable sentences mentioned the conclusion of our work in “Related work” section (line 91-92) and “Inserting Scalable BN layer to improve the classification performance of Softmax loss function” section (line 189-190, 198-200). - Response to comment: (The configurations evaluated in the table to compare results should be also discussed)
Response: We added explanation to why these parameters and the function of these parameters in the final part of “Experiment and result” section (line 221-226). “Here we set two sets hyperparameter s=7.5, m=0.35 and s=10, m=0.5, which is following the original PEDCC-loss [9]. The hyperparameter s is used to let the network trainable. Because when the features and every column of weight are normalized, the output range is only [-1, 1], which is not suitable for Softmax loss [15]. Another hyperparameter m represents the expected cosine margin between each class after training [8].” - Response to comments in the attached document:
(1) We are very sorry for our incorrect writing. Now the parameters in our paper are written in italics. And some other errors in line 61, 76, 82, 87, 94, 139-140, 155-158, 162, 166, 180, 183, 188-191, 209-213, 215-216, 219-220.
(2) In section 2.2, we changed the informal sentences (line 109-110), and added a reference to support the sentence “Experience has proved that training will be more stable and convergent if each channel of RGB image is normalized [12]” (line 112). The very long sentence is re-wrote to make it clearer. Besides, we explained more about why batch normalization can improve the training phase of neural network.
(3) For the sentence “except let the linear layer’s input feature be more effective, simply multiplying the value of the linear layer weight to a constant value greater than 1, the same result will be obtained.”
This sentence is modified (line 147). The effectiveness of the feature is determined by the training process, which is finally reflected in the classification accuracy.
(4) It seems that the part including Figure 2 and 3 (line 181-197) are appropriate in section 3.
This sentence is in the section 3 (Inserting Scalable BN…). If it still needs to be changed please kindly tell us.
(5) We are very sorry for the overlap sentences. The sentences (line 189-190) are deleted.
(6) The title of section 4 is changed and the comments of result part are responsed above.
Finally, special thanks to you for your time and good comments. That means a lot to us.
Round 2
Reviewer 1 Report
All concerns are met.